# The Heteroepitaxy of Thick *β*-Ga_2_O_3_ Film on Sapphire Substrate with a *β*-(Al_x_Ga_1−x_)_2_O_3_ Intermediate Buffer Layer

**DOI:** 10.3390/ma16072775

**Published:** 2023-03-30

**Authors:** Wenhui Zhang, Hezhi Zhang, Song Zhang, Zishi Wang, Litao Liu, Qi Zhang, Xibing Hu, Hongwei Liang

**Affiliations:** 1School of Microeletronics, Dalian University of Technology, Dalian 116024, China; 2Jiangsu Xinguanglian Technology Company Co., Ltd., Wuxi 214192, China; 3The 46th Research Institute of China Electronics Technology Group Corporation, Tianjin 300220, China

**Keywords:** *β*-(Al_x_Ga_1−x_)_2_O_3_ buffer layer, *β*-Ga_2_O_3_ thick film, heteroepitaxy

## Abstract

A high aluminum (Al) content *β*-(Al_x_Ga_1−x_)_2_O_3_ film was synthesized on c-plane sapphire substrate using the gallium (Ga) diffusion method. The obtained *β*-(Al_x_Ga_1−x_)_2_O_3_ film had an average thickness of 750 nm and a surface roughness of 2.10 nm. Secondary ion mass spectrometry results indicated the homogenous distribution of Al components in the film. The Al compositions in the *β*-(Al_x_Ga_1−x_)_2_O_3_ film, as estimated by X-ray diffraction, were close to those estimated by X-ray photoelectron spectroscopy, at ~62% and ~61.5%, respectively. The bandgap of the *β*-(Al_x_Ga_1−x_)_2_O_3_ film, extracted from the O 1s core-level spectra, was approximately 6.0 ± 0.1 eV. After synthesizing the *β*-(Al_x_Ga_1−x_)_2_O_3_ film, a thick *β*-Ga_2_O_3_ film was further deposited on sapphire substrate using carbothermal reduction and halide vapor phase epitaxy. The *β*-Ga_2_O_3_ thick film, grown on a sapphire substrate with a *β*-(Al_x_Ga_1−x_)_2_O_3_ buffer layer, exhibited improved crystal orientation along the (-201) plane. Moreover, the scanning electron microscopy revealed that the surface quality of the *β*-Ga_2_O_3_ thick film on sapphire substrate with a *β*-(Al_x_Ga_1−x_)_2_O_3_ intermediate buffer layer was significantly improved, with an obvious transition from grain island-like morphology to 2D continuous growth, and a reduction in surface roughness to less than 10 nm.

## 1. Introduction

*β*-Ga_2_O_3_ has garnered significant attention in recent years due to its wide bandgap, high breakdown electrical field, and Baliga’s figure of merit [1,2]. The synthesis of *β*-Ga_2_O_3_ films on various substrates, such as Ga_2_O_3_ [3,4], sapphire [5,6], and GaAs [7], has been widely reported. Among these substrates, homoepitaxial *β*-Ga_2_O_3_ films exhibit a smooth surface without cracks or dislocations, making it the most favorable substrate currently available. However, the cost of *β*-Ga_2_O_3_ substrate is relatively high, which limits its market application. Compared with *β*-Ga_2_O_3_ substrate, sapphire is an economic and cost-effective substrate for heteroepitaxy. Several groups have attempted to synthesize *β*-Ga_2_O_3_ on sapphire substrate using various methods, including pulsed layer deposition (PLD) [8], molecular beam epitaxy (MBE) [4,9], halide vapor phase epitaxy (HVPE) [3,10], carbothermal reduction [11], metal organic chemical vapor deposition (MOCVD) [12,13], and low-pressure chemical vapor deposition (LPCVD) [14]. PLD and MBE, allowing for precision controllability, are able to achieve high-quality *β*-Ga_2_O_3_ films on sapphire. Nevertheless, their growth rate is not sufficient for fast growth of thick films. HVPE, carbothermal reduction, and MOCVD methods show competitive growth rates over 5 μm/h. Among these, carbothermal reduction is a promising technique for the growth of thick *β*-Ga_2_O_3_ films, as it avoids the use of corrosive precursor gases. However, the crystal quality is significantly degraded under fast growth because of the lattice mismatch between the corundum structure of sapphire (α-Al_2_O_3_) and the monoclinic structure of *β*-Ga_2_O_3_. One solution for heteroepitaxy on sapphire is to use a larger bandgap material as a buffer layer to mitigate the lattice mismatch-induced strain, as is commonly known for growth of GaN on AlN/sapphire template. In the case of *β*-Ga_2_O_3_, we employed the ((Al_x_Ga_1−x_)_2_O_3_) film as a buffer layer to indirectly mitigate the lattice mismatch. Y. Cheng et al. grew *β*-(Al_x_Ga_1−x_)_2_O_3_ on a sapphire substrate and used it as an intermediate buffer layer to epitaxially grow *β*-Ga_2_O_3_ thin films. The results showed that *β*-(Al_x_Ga_1−x_)_2_O_3_, as an intermediate buffer layer, can reduce lattice mismatch and improve the crystal quality of *β*-Ga_2_O_3_ thin films [15].

Currently, molecular beam epitaxy (MBE) [16], pulsed laser deposition (PLD) [17], and metal–organic vapor deposition (MOCVD) [18] are common methods used to grow (Al_x_Ga_1−x_)_2_O_3_ alloys. However, achieving a growth strategy for *β*-(Al_x_Ga_1−x_)_2_O_3_ film remains a challenge. This obstacle is associated with synthesizing a stable high-Al-content *β*-(Al_x_Ga_1−x_)_2_O_3_ by direct growth techniques, such as MOCVD, MBE, and PLD, due to the phase transformation from *β* to γ for (Al_x_Ga_1−x_)_2_O_3_ when the Al content exceeds 30% [19,20]. Additionally, these methods are based on expensive vacuum equipment. Instead of direct epitaxial *β*-(Al_x_Ga_1−x_)_2_O_3_ on sapphire, we utilized the gallium (Ga) diffusion method, which has previously been used to fabricate Ga-diffused waveguides in sapphire [21].

In this paper, the *β*-(Al_x_Ga_1−x_)_2_O_3_ film was grown on c-plane sapphire substrate in a high-temperature tubular furnace by the gallium (Ga) diffusion method, serving as an intermediate buffer layer for the subsequent heteroepitaxial growth of the thick *β*-Ga_2_O_3_ film. The obtained *β*-(Al_x_Ga_1−x_)_2_O_3_ film displayed a high crystal quality, with a thickness of approximately 750 nm. The distribution of Al components in the film was homogenous, with an Al content of approximately 62%. After synthesizing the *β*-(Al_x_Ga_1−x_)_2_O_3_ buffer layer, the *β*-Ga_2_O_3_ thick film was further deposited on the *β*-(Al_x_Ga_1−x_)_2_O_3_/sapphire template using two methods: carbothermal reduction, reported recently by our group [11], and HVPE. Finally, we characterized the properties of the *β*-Ga_2_O_3_ thick film on sapphire with and without the *β*-(Al_x_Ga_1−x_)_2_O_3_ buffer layer. The results showed that the *β*-Ga_2_O_3_ thick film, grown on a sapphire substrate with a *β*-(Al_x_Ga_1−x_)_2_O_3_ buffer layer, improved crystal orientation and surface quality.

## 2. Experiments

The (Al_x_Ga_1−x_)_2_O_3_ film was synthesized using a high-temperature tubular furnace, as illustrated in Figure 1. Ga2O3 powder with purity of 99.999% as the source material was put in a corundum crucible. The sapphire substrate was inserted into the corundum crucible, and the system was subjected to setting temperature of 1450 °C. The growth process took place in an anoxic atmosphere, where 1.5 slm argon (Ar) was maintained at the pressure of 3 × 10^4^ Pa for 2 h. In a neutral gas atmosphere, Ga_2_O_3_ powder underwent decomposition into volatile Ga_2_O(g), which further decomposed into gaseous Ga, as illustrated below:(1)$$Ga_{2}O_{3}\left(s\right)\to Ga_{2}O\left(g\right)+O_{2}\left(g\right)$$
(2)$$Ga_{2}O\left(g\right)\to 2Ga\left(g\right)+1/2O_{2}\left(g\right)$$

The Ga species diffused into α-Al_2_O_3_, resulting in the formation of a *β*-(Al_x_Ga_1−x_)_2_O_3_ buffer layer according to the Al_2_O_3_-Ga_2_O_3_ phase diagram.

The Ga_2_O_3_ thick film was grown using the carbothermal reduction method in a home-made growing system as shown in Figure 2. During the growth process, 20 sccm of O_2_ and 500 sccm of Ar were kept for 2 h at a pressure of 3 × 10^4^ Pa. The growth condition for HVPE was as follows: the ratio of flow rate between HCl and O_2_, setting growth temperature, and pressure were 10/30, 1060 °C, and 5 × 10^4^ Pa, respectively. The growth was kept for 2 h for HVPE. 

## 3. Results and Discussions

The thickness of the (Al_x_Ga_1−x_)_2_O_3_ film was investigated using cross-sectional scanning electron microscopy (FEI Nova Nano SEM 450), as shown in Figure 3a, which presents a clear interface between (Al_x_Ga_1−x_)_2_O_3_ and sapphire substrate. The *β*-(Al_x_Ga_1−x_)_2_O_3_ film thickness was measured to be 750 nm, corresponding to a growth rate of 375 nm/h. The surface morphology and roughness of the (Al_x_Ga_1−x_)_2_O_3_ film were characterized by top-view scanning electron microscopy and atomic force microscopy (AFM, Dimension Icon, Bruker, Germany). Figure 3b shows the equilateral triangular morphology of the (Al_x_Ga_1−x_)_2_O_3_ film, which corresponds to the arrangement of oxygen atoms (equilateral triangles) on a c-plane sapphire substrate. When *β*-Ga_2_O_3_ is grown on a c-plane sapphire substrate, the oxygen atoms on the surface between the *β*-Ga_2_O_3_ (-201) plane and the c-plane sapphire are arranged in equilateral triangles, leading to (-201)-oriented growth. The AFM image corroborates the equilateral triangular morphology and corresponds to a root mean square roughness (RMS) of around 2.10 nm.

The crystalline orientation was characterized using high-resolution X-ray diffraction (HRXRD, Bruker D8 Advance). Figure 4a depicts a θ–2θ scan XRD result, showing three diffraction peaks at 19.3°, 39.12°, and 60.2°, corresponding to the (-201), (-402), and (-603) planes of monoclinic *β*-(Al_x_Ga_1−x_)_2_O_3_, respectively. The growth of *β*-(Al_x_Ga_1−x_)_2_O_3_ film on (0001) sapphire substrate along the (-201) crystal plane is attribute to the similar arrangement of oxygen atoms(equilateral triangles) at the surface monoclinic structure *β*-Ga_2_O_3_ (-201) plane and corundum structure α-Al_2_O_3_ (0001) plane. The diffraction peaks of *β*-Ga_2_O_3_ (PDF#43-1012) were used as a standard. As shown in Table 1, the diffraction peak position of *β*-(Al_x_Ga_1−x_)_2_O_3_ film shifted to a higher diffraction angle. This phenomenon arose because the Ga^3+^ ion was replaced by the smaller-radius Al^3+^ ion, causing the lattice spacing to shrink and the diffraction peak to move to a higher angle. Considering the monoclinic structure, the Al compositions in *β*-(Al_x_Ga_1−x_)_2_O_3_ film was determined using following expression [22,23].
(3)$$\frac{1}{d^{2}}=\frac{h^{2}}{a^{2}sin^{2}\beta }+\frac{k^{2}}{b^{2}}+\frac{l^{2}}{c^{2}sin^{2}\beta }-\frac{2hl\cos\beta }{acsin^{2}\beta }$$
where *h* = −4, *k* = 0, and *l* = 2. Based on the (-402) diffraction peak position and Equation (3), we obtained the Al compositions in films to be around 62%. The crystalline quality of *β*-(Al_x_Ga_1−x_)_2_O_3_ film was characterized by ω rocking curve spectra. As shown in Figure 4b, the full width at half maximum (FWHM) of the (-201) plane was around 0.42°, indicating that the *β*-(Al_x_Ga_1−x_)_2_O_3_ film had high crystalline quality, although the crystalline quality of *β*-(Al_x_Ga_1−x_)_2_O_3_ films prepared by PLD, MBE, and MOVD deteriorated when the Al composition exceeded 30% [19,20].

The content of Al in *β*-(Al_x_Ga_1−x_)_2_O_3_ films was further determined by X-ray photoelectron spectroscopy (XPS, K-alpha+). A wide survey spectrum clearly showed peaks of Al 2s and Al 2p for the *β*-(Al_x_Ga_1−x_)_2_O_3_ film, along with *β*-Ga_2_O_3_ crystal as a reference, as shown in Figure 5a. The Al compositions in the film were estimated from the Al 2p and Ga 2p core-level peak areas, considering the sensitivity factors of the elements. Figure 5b,c show the Al 2p and Ga 2p core-level spectra for the film. The Al 2p peak in the *β*-(Al_x_Ga_1−x_)_2_O_3_ film displayed a binding energy of 73.96 eV. This shift towards a lower binding energy compared with the Al 2p peak in the sapphire substrate (74.5 eV) can be attributed to the formation of Al-O-Ga bonds [24]. Based on the XPS results, the Al composition in the *β*-(Al_x_Ga_1−x_)_2_O_3_ film was around 61.5%, which is consistent with the quantitative results obtained from the XRD analysis.

The bandgap of *β*-(Al_x_Ga_1−x_)_2_O_3_ film was determined by analyzing the O 1s core-level spectra in XPS. This approach has been established by previous studies [25,26,27,28]. The bandgap energy can be derived from the difference between the core-level peak energy and the initial inelastic losses [29]. Figure 6 presents the O1s core-level spectra obtained from XPS analysis of the *β*-(Al_x_Ga_1−x_)_2_O_3_ film. The O1s peak energy was 530.8 eV, while the initial inelastic losses were 536.8 eV. The bandgap of *β*-(Al_x_Ga_1−x_)_2_O_3_ film extracted from the O1s spectra was approximately 6.0 ± 0.1 eV, which is consistent with the bandgap energy estimated by absorption spectra in previous literature [30]. The impurity in the *β*-(AlxGa1-x)2O3 film was investigated using time-of-flight secondary ion mass spectrometry (SIMS, IONTOF 5). Figure 7 shows the TOF-SIMS depth profile for the *β*-(Al_x_Ga_1−x_)_2_O_3_ film, which revealed that the impurity in the film was negligible. Additionally, the *β*-(Al_x_Ga_1−x_)_2_O_3_ film exhibited a homogenous distribution of Al.

High-resolution field emission transmission electron microscopy (HRTEM, JEM F200) was used to investigate the interface microstructure of *β*-(Al_x_Ga_1−x_)_2_O_3_/Al_2_O_3_ heterojunction, as shown in Figure 8. The crystal lattice spacing of *β*-(Al_x_Ga_1−x_)_2_O_3_ was 0.456 nm, corresponding to the (-201) crystal plane spacing. The transition layer thickness in the vicinity of the interface on sapphire substrates was approximately ~3.3 nm. As shown in the inset, the fast Fourier transform (FFT) diffraction patterns describe the monoclinic *β*-phase crystal structure and the high crystalline quality of the *β*-(Al_x_Ga_1−x_)_2_O_3_ film.

After preparing the buffer layer, the *β*-(Al_x_Ga_1−x_)_2_O_3_/sapphire template was transferred to a fast epitaxial *β*-Ga_2_O_3_ thick film through carbothermal reduction and HVPE techniques, respectively. A reference sample of *β*-Ga_2_O_3_ on sapphire without a buffer layer was also grown under the same conditions for comparison. The growth rate for all samples was approximately 4~6 μm/h, depending on the film thickness measured by cross-sectional SEM image as shown in Appendix A. Figure 9 presents the θ–2θ scan XRD characterization of *β*-Ga_2_O_3_ thick film on sapphire with and without a *β*-(Al_x_Ga_1−x_)_2_O_3_ buffer layer, respectively. The samples grown on the buffer layer showed a clear appearance of dominant (-201) and high-order *β*-Ga_2_O_3_ diffraction peaks, as shown in Figure 9b,c for the growth carried out by carbothermal reduction and HVPE, respectively. However, the *β*-Ga_2_O_3_ on sapphire without a buffer layer revealed a competitive crystal orientation of (400), (002), (-403), and (-313) peaks marked in Figure 9a, except for the peak of (-201) orientation planes. This competitive crystal orientation is due to the lattice mismatch related anisotropic growth, which leads to the existence of rhombic prism faces, as marked in the rectangle area indicated in Figure 10a. The XRD results demonstrate that miscellaneous crystalline facets were strongly inhibited for *β*-Ga_2_O_3_ thick film on sapphire by means of a *β*-(Al_x_Ga_1−x_)_2_O_3_ buffer layer, which implies a much-improved crystalline quality.

The surface properties of *β*-Ga_2_O_3_ thick film were investigated by SEM. As shown in Figure 10a, the surface morphology of *β*-Ga_2_O_3_ on sapphire without a buffer layer displayed the pseudo hexagonal shape with an average grain size of 4 μm and well-defined boundaries. The details of coalesced *β*-Ga_2_O_3_ grain were composed of rhombic prism faces, which contribute to the sub-peak in XRD measurement as we mentioned above. Taking into account the *β*-Ga_2_O_3_ thick film on sapphire with *β*-(Al_x_Ga_1−x_)_2_O_3_ buffer layer, the SEM image explicitly unveiled a significant improvement in surface quality through the obvious transition from grain island-like morphology to 2D continuous growth, as shown in Figure 10b,c, respectively.

The surface roughness was finally identified by AFM measurements of 5 × 5 μm. The AFM results displayed highly correlated morphology with the high magnification SEM images for all three samples. The RMS surface roughness was 74 nm for the *β*-Ga_2_O_3_ thick film on sapphire without a buffer layer, as shown in Figure 11a. By contrast, the use of the *β*-(Al_1−x_Ga_x_)_2_O_3_ buffer layer resulted in a much smoother surface, as confirmed by the RMS values of 9 nm and 5 nm for the epitaxial film prepared by carbothermal reduction and HVPE, respectively, as shown in Figure 11b,c. Further effort would be to optimize the growth parameter to obtain an even smoother surface without additional pits. 

## 4. Conclusions

In this paper, we described the heteroepitaxy of a thick *β*-Ga_2_O_3_ film on c-plane sapphire substrate employing a larger bandgap *β*-(Al_x_Ga_1−x_)_2_O_3_ buffer layer to improve the growth quality while maintaining a comparable growth rate (~5 μm/h). We used the gallium (Ga) diffusion method to create a *β*-(Al_x_Ga_1−x_)_2_O_3_ buffer layer on c-plane sapphire substrate. The Al composition in the *β*-(Al_x_Ga_1−x_)_2_O_3_ film estimated by XRD was ~62%, which is comparable to the result of ~61.5% estimated by XPS. The bandgap of the *β*-(Al_x_Ga_1−x_)_2_O_3_ film derived from the O 1s core-level spectra was around 6.0 ± 0.1 eV. SIMS results indicated a homogenous distribution of the Al element in the film. After the formation of the *β*-(Al_x_Ga_1−x_)_2_O_3_ buffer layer, the *β*-Ga_2_O_3_ thick film was deposited on the *β*-(Al_x_Ga_1−x_)_2_O_3_/sapphire template by carbothermal reduction and HVPE, respectively. The *β*-Ga_2_O_3_ thick film on the sapphire substrate with the *β*-(Al_x_Ga_1−x_)_2_O_3_ buffer layer exhibited an evident appearance of dominated (-201) crystalline facets and inhibited (400), (002), (-403), and (-313) miscellaneous crystalline facets, regardless of the growth method. The surface quality of the *β*-Ga_2_O_3_ thick film on the *β*-(Al_x_Ga_1−x_)_2_O_3_/sapphire template was significantly improved compared with that without a buffer layer, as measured by SEM and AFM, with the obvious transition from a grain island-like morphology to 2D continuous growth and a reduction of surface roughness to less than 10 nm.

## Figures and Tables

**Figure 1 materials-16-02775-f001:**
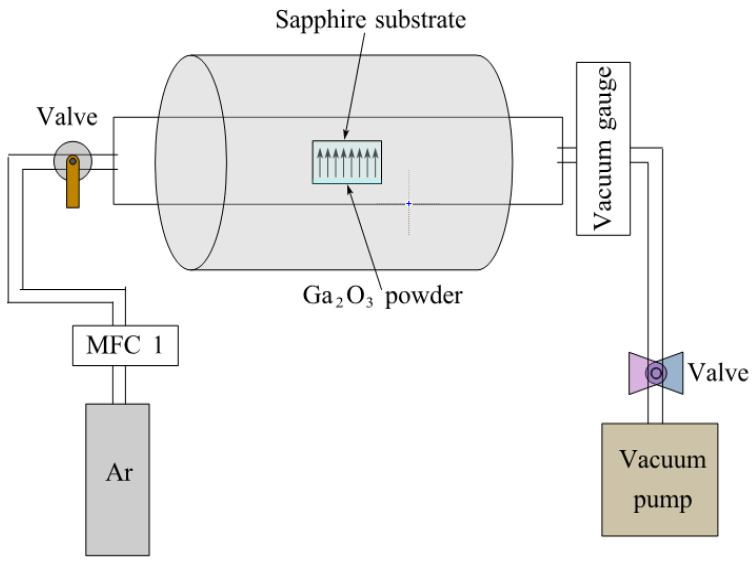
The schematic diagram of the high-temperature tubular furnace.

**Figure 2 materials-16-02775-f002:**
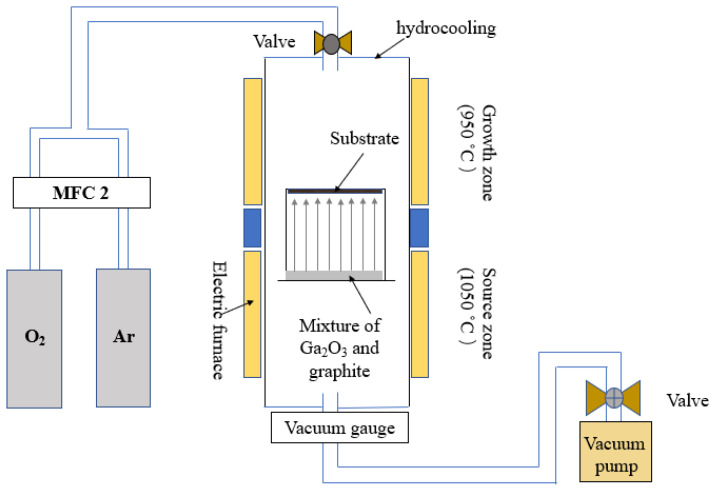
The schematic diagram of the home-made growing system.

**Figure 3 materials-16-02775-f003:**
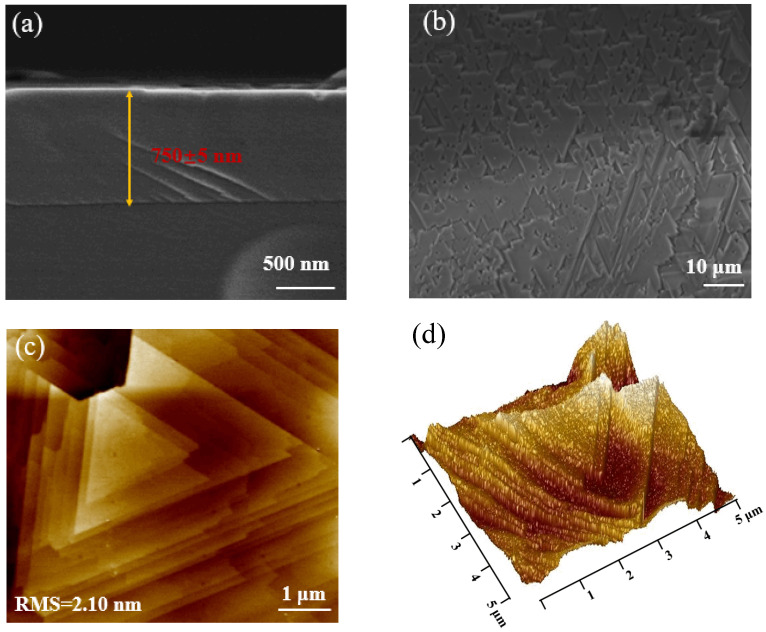
The cross-sectional SEM image of *β*-(Al_x_Ga_1−x_)_2_O_3_ film grown on sapphire substrate (**a**). Corresponding surface SEM image (**b**) and 2D (**c**) and 3D (**d**) AFM images.

**Figure 4 materials-16-02775-f004:**
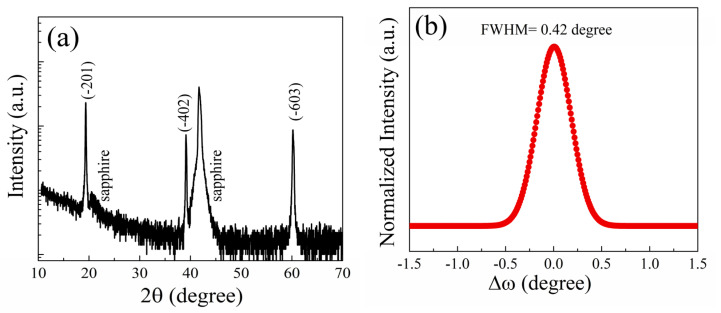
The XRD result of the *β*-(Al_x_Ga_1−x_)_2_O_3_ film grown on sapphire substrate (**a**) and the ω rocking curve of (-201) plane for the film (**b**).

**Figure 5 materials-16-02775-f005:**
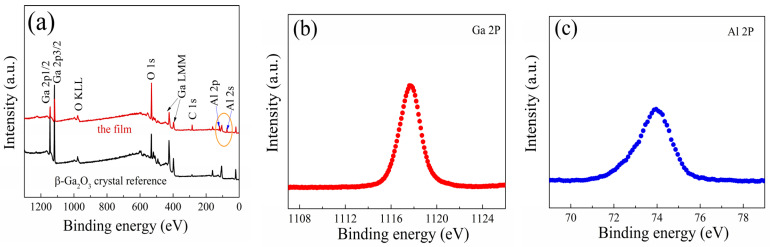
The XPS wide survey spectra of *β*-(Al_x_Ga_1−x_)_2_O_3_ film on sapphire substrate (**a**). The Al 2p core-level spectra (**b**) and Ga 2p core-level spectra (**c**).

**Figure 6 materials-16-02775-f006:**
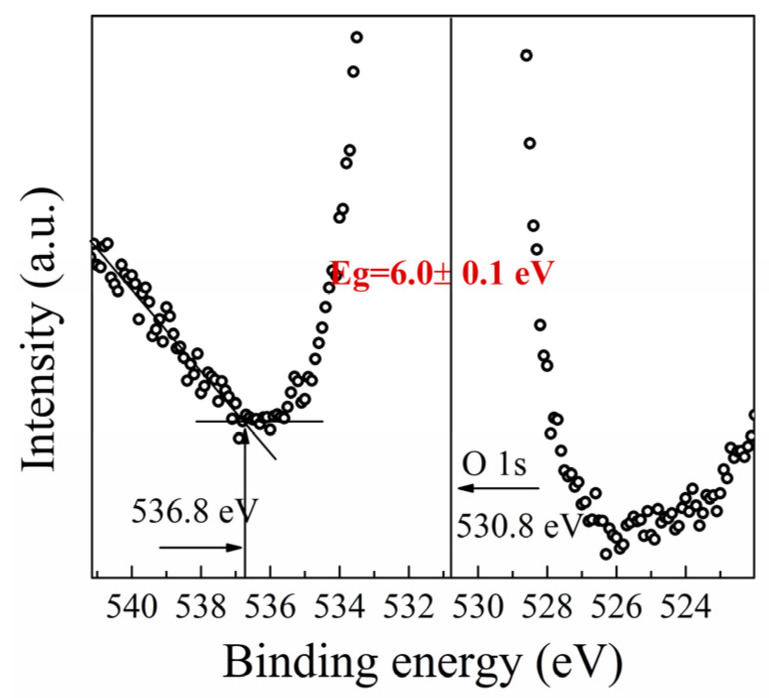
The peak energy and the inelastic losses of O 1s for the *β*-(Al_x_Ga_1−x_)_2_O_3_ film.

**Figure 7 materials-16-02775-f007:**
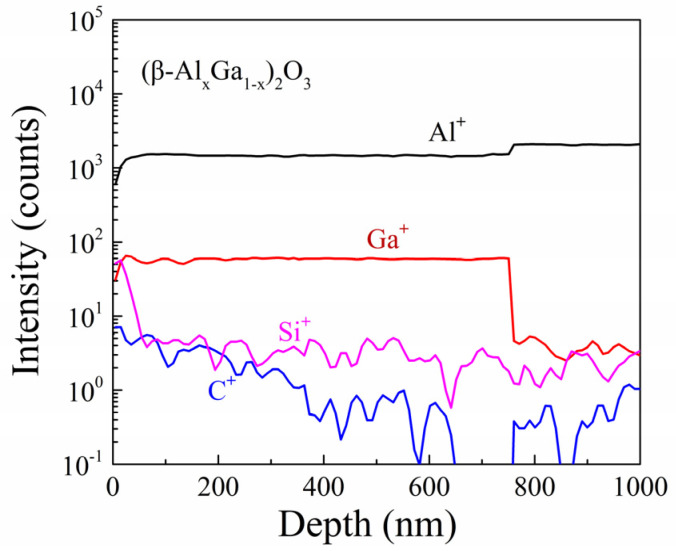
The TOF-SIMS depth profile of *β*-(Al_x_Ga_1−x_)_2_O_3_ film.

**Figure 8 materials-16-02775-f008:**
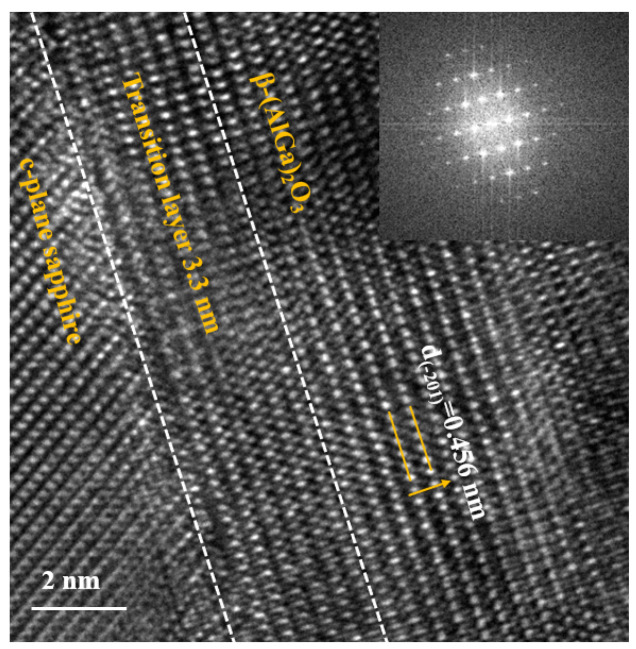
The HRTEM image of *β*-(Al_x_Ga_1−x_)_2_O_3_ film on sapphire substrate.

**Figure 9 materials-16-02775-f009:**
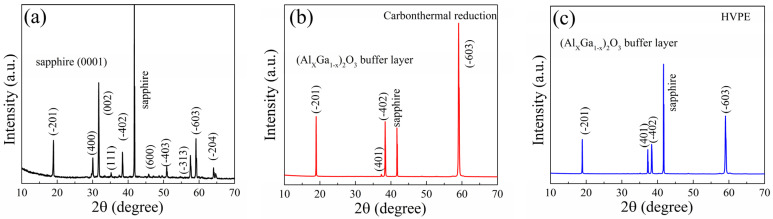
The XRD result of *β*-Ga_2_O_3_ thick film grown directly on sapphire substrate (**a**). The XRD result of *β*-Ga_2_O_3_ thick film grown on *β*-(Al_x_Ga_1−x_)_2_O_3_/sapphire substrate by carbothermal reduction (**b**) and HVPE (**c**) methods.

**Figure 10 materials-16-02775-f010:**
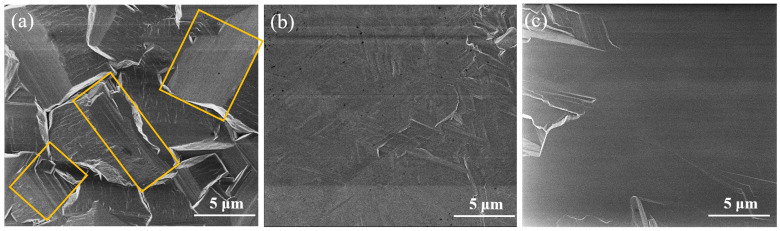
The surface SEM image of *β*-Ga_2_O_3_ thick film grown directly on sapphire substrate (**a**). The surface SEM image of *β*-Ga_2_O_3_ thick film grown on *β*-(Al_x_Ga_1−x_)_2_O_3_/sapphire substrate by carbothermal reduction (**b**) and HVPE (**c**) methods.

**Figure 11 materials-16-02775-f011:**
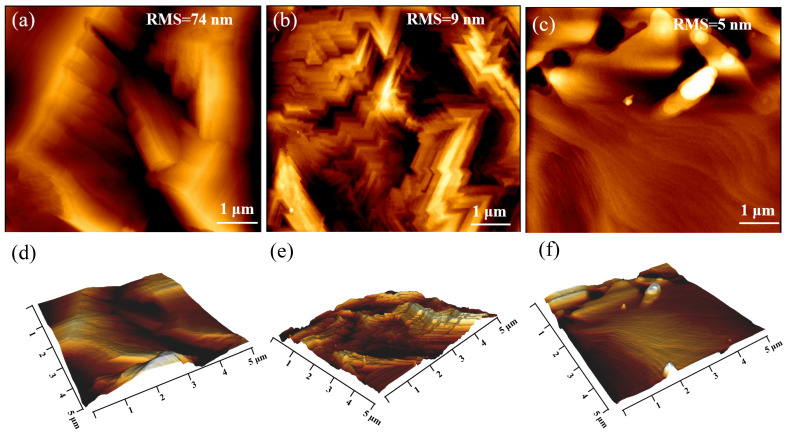
The 2D (**a**) and 3D (**d**) AFM result of *β*-Ga_2_O_3_ thick film grown directly on sapphire substrate. The 2D and 3D AFM results of *β*-Ga_2_O_3_ thick film grown on *β*-(Al_x_Ga_1−x_)_2_O_3_/sapphire substrate by carbothermal reduction (**b**,**e**) and HVPE (**c**,**f**).

**Table 1 materials-16-02775-t001:** The diffraction peak positions of *β*-(Al_x_Ga_1−x_)_2_O_3_ film and the corresponding *β*-Ga_2_O_3_ standard diffraction peak positions.

Sample	(-201)	(-402)	(-603)
*β*-Ga_2_O_3_ (reference)	18.95	38.404	59.19
(Al_x_Ga_1−x_)_2_O_3_	19.3	39.12	60.2

## Data Availability

The data that support the findings of this study are available from the corresponding author upon reasonable request.

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
