# Peer review of "The Heteroepitaxy of Thick β-Ga2O3 Film on Sapphire Substrate with a β-(AlxGa1−x)2O3 Intermediate Buffer Layer"

_materials, 2023, doi:10.3390/ma16072775_

Round 1

Reviewer 1 Report

The effect of the interface layer of β-(AlxGa1-x)2O3 is mainly due to the interaction between the sapphire substrate (Al2O3) and the deposited β-Ga2O3 layer. In addition β-(AlxGa1-x)2O3 deposited layer can raise more attention. The β-Ga2O3 layer is essential for optoelectronics and electronic applications due to its wide bandgap behavior.
The topic is of great importance as it helps to gain a better understanding of the physics at the interfaces. Additionally, the root mean square (RMS) factor is critical for practical applications. Hence, the authors have considered this factor through different diffused Al atoms.

The authors have delved into understanding the impact of using a buffer layer of β-(AlxGa1-x)2O3. As the substrate is already Al2O3, introducing more Al in the β-(AlxGa1-x)2O3 will make the interface smoother and reduce the internal strain.

The study could investigate the effect of different concentrations of Al and the thickness of the β-(AlxGa1-x)2O3 layer.

The conclusion of the research article is well-written and summarizes the primary findings.

I suggest that the literature review should be expanded and compared with the results of this study.

Lastly, I would like to request that the authors provide 3D images of the atomic force microscopy (AFM) results. Such images are more helpful than the 2D ones.

-          In fig 2(a). how did you define the error values of 5?

-          In fig 22, about the RSM, is it a whole image, or a horizontal or vertical line? because I see that the RMS value is a bit low compared to the SEM image morphology

-          May you enhance the literature review and comparing with your findings

-          Please check the grammar and the writing types of the whole paper 

Reviewer 2 Report

The paper “The heteroepitaxy of thick β-Ga2O3 film on Sapphire substrate with β-(AlxGa1-x)2O3 intermediate buffer layer” presented by Wenhui Zhang and coworkers, discusses new cost- and time-effective experimental technique of epitaxially growth of β- Ga2O3 film. The authors propose to use a buffer layer obtained by diffusing Ga into a sapphire substrate to improve the quality of β-Ga2O3 film grown with carbothermal reduction and/or HVPE. With help of SEM, HRTEM, AFM, XRD, XPS, SIMS, the success of the proposed approach has been clearly demonstrated. The article is well written, structured and organized. Presented results contains useful information for readers who are interested in epitaxy, chemical synthesis methods and wide gap semiconductors. In my opinion, the paper meets the requirements of Materials journal. I recommend to accept, but before publication, one shortcoming of the manuscript should be rectified.

Because of main article result is related to technology, additional information at Experiments part is necessary. More details about carbothermal reduction and HVPE processes are needed. I think, additional schematic diagram of homemade vertical furnace, for example depicted on fig.1b, will help readers clearly understand the results presented in paper. Also, I recommend to indicate operating temperatures on scheme.

Round 2

Reviewer 1 Report

Thank you for your report